# Characterization of Parthanatos in Breast Cancer: Implications for Prognosis and PARP Inhibitor Resistance

**DOI:** 10.3390/bioengineering12060586

**Published:** 2025-05-29

**Authors:** Junjie Tang, Qian Liu, Wei Du, Linxi Chen, Feiyang Qi, Ranxin Zhang, Bang H. Hoang, David S. Geller, Rui Yang, Jichuan Wang, Li Hu

**Affiliations:** 1Key Laboratory of Carcinogenesis and Translational Research (Ministry of Education), Familial & Hereditary Cancer Center, Peking University Cancer Hospital & Institute, Beijing 100142, China; tjj_njmu@163.com (J.T.); qianliu0221@163.com (Q.L.); 2The First Clinical Medical School, Nanjing Medical University, Nanjing 211166, China; 3Breast Center, Peking University People’s Hospital, Beijing 100044, China; lilacdw301@163.com; 4Musculoskleletal Tumor Center, Beijing Key Laboratory for Musculoskeletal Tumors, Peking University People’s Hospital, Beijing 100041, China; shining0xi@163.com (L.C.); 1910301241@pku.edu.cn (F.Q.); 5Department of Orthopedic Surgery, Montefiore Medical Center, Albert Einstein College of Medicine, Bronx, NY 10467, USA; ranxin.zhang@einsteinmed.edu (R.Z.); bahoang@montefiore.org (B.H.H.); dgeller@montefiore.org (D.S.G.); rui.yang@uth.tmc.edu (R.Y.)

**Keywords:** parthanatos, breast cancer, tumor immune microenvironment, single-cell sequencing, subtypes

## Abstract

Parthanatos, a novel form of programmed cell death mediated by PARP1 and driven by DNA damage, has not been comprehensively characterized in breast cancer (BC). Given the widespread clinical use of PARP1 inhibitors for treating BRCA-mutant breast cancers, understanding the role of parthanatos is crucial. In this study, we systematically analyzed the role and clinical significance of parthanatos in BC by integrating genomic, transcriptomic, and clinical data. Our analysis revealed significant differential expression of parthanatos-related genes between BC and normal breast tissues, with frequent copy number variations (CNVs) affecting gene expression. At the single-cell level, parthanatos-related genes were primarily expressed in breast cancer cells and macrophages, and genomic instability scores were positively correlated with parthanatos pathway scores. Unsupervised clustering identified two BC subtypes based on parthanatos-related gene expression: C1 (parthanatos-high) and C2 (parthanatos-low). C1 exhibited a poorer prognosis, reduced immune infiltration, and potential resistance to PARP inhibitors compared to C2. This study represents the first comprehensive investigation of parthanatos in BC, offering insights into its role in tumor progression and highlighting its potential link to PARP inhibitor resistance and poor clinical outcomes.

## 1. Introduction

Parthanatos, a novel form of programmed cell death (PCD) mediated by PARP1 and driven by DNA damage, is a regulated form of cell death that has not been comprehensively characterized in BC. PCD encompasses apoptosis, autophagy-dependent cell death, necroptosis, pyroptosis, ferroptosis, and others, all orchestrated by biomacromolecules [1,2]. Increasing evidence suggests that PCD mechanisms play key roles in tumorigenesis and may serve as foundations for novel therapeutic strategies. Given the widespread clinical use of PARP1 inhibitors for treating *BRCA*-mutant BC, understanding the role of parthanatos is crucial [1,3].

Notably, parthanatos involves three key phases: activation of PARP1, translocation of apoptosis-inducing factor (AIF), and DNA fragmentation, ultimately leading to cell death [1,4]. PARP1 activation is driven by DNA damage, and its effects vary depending on the severity of the damage. Mild DNA damage prompts PARP1 activation to facilitate DNA repair via base excision repair (BER), and extensive DNA damage results in the overactivation of PARP1, leading to parthanatos [4]. Increased parthanatos activity has been observed in ovarian and lung cancers compared to normal tissues, with PARP1 overexpression at both mRNA and protein levels in gastric and small-cell lung cancers, correlating with poor prognosis [5]. Additionally, PARP1 knockout mice exhibit reduced tumor growth, particularly in pancreatic and colorectal cancers, underscoring the association between parthanatos and cancer progression [6,7].

Breast cancer may be strongly associated with parthanatos. Key genes in the parthanatos pathway, such as PARP1, PTEN, and AIFM1, are critical drivers of breast cancer initiation, progression, drug resistance, and metastasis [8,9]. PARP1 is significantly amplified and highly expressed in recurrent breast cancer, while PTEN loss has been linked to an increased risk of breast cancer, and AIFM1 plays a crucial role in tumor invasion and metastasis by inactivating PTEN [10,11]. Moreover, PARP inhibitors (PARPi) are widely used to exploit synthetic lethality in conjunction with *BRCA* mutations, serving as an important treatment strategy for BC patients harboring these mutations [12]. Despite these associations, parthanatos remains underexplored in breast cancer, and its role in tumor initiation, progression, and clinical relevance requires further investigation.

To address this gap, we conducted a comprehensive analysis of parthanatos and evaluated its roles in breast tumorigenesis and prognostic significance, particularly its relationship with the response to PARP inhibitors by using multi-omics data, including bulk RNA-seq, single-cell RNA-seq, and whole-exome sequencing (WES), obtained from various public databases. First, we investigated the mRNA expression characteristics of parthanatos-related genes at both bulk and single-cell levels in breast cancers. Next, we analyzed the genomic alterations of these parthanatos-related genes in breast cancer, followed by analyzing the correlation between genomic alterations and mRNA expression level. Finally, based on the expression characteristics of parthanatos-related genes, we performed unsupervised clustering to classify breast cancer into distinct subtypes and compared the clinical characteristics, prognosis, and immune microenvironment between these subtypes. Using real-world PARP inhibitor treatment cohorts, we further explored the association between parthanatos-related subtypes and the response to PARP inhibitors in breast cancer. This comprehensive investigation provides new insights into parthanatos in BC, highlighting its potential role in tumor progression and resistance to PARP inhibitors.

## 2. Methods

### 2.1. Sample Source and Parthanatos-Related Gene Source

We downloaded multi-omics data from The Cancer Genome Atlas (TCGA) regarding breast cancer, including CNVs, single nucleotide variants (SNV), and mRNA expression profiles, along with corresponding clinical information. After filtering out missing values, a total of 1082 breast cancer patients were included in the analysis. Clinical characteristics and NGS data were obtained from the cBioPortal database (https://www.cbioportal.org, accessed on 1 February 2025) or original publications.

The single-cell breast cancer dataset was sourced from GSE161529, profiling 34 treatment-naive primary tumors, encompassing 13 estrogen receptor and/or progesterone receptor positive (ER/PR+), 6 human epidermal growth factor receptor 2 positive (HER2+), and 8 triple-negative breast cancers (TNBC) [13].

The I-SPY2 study is an ongoing open-label, multicenter, adaptively randomized phase 2 platform trial for high-risk, stage II/III breast cancer, evaluating multiple investigational arms in parallel. We downloaded the expression data and corresponding clinic data of the PARP inhibitor arm (*N* = 71) to assess the response of parthanatos-related breast cancer subtypes to PARP inhibitors [14].

Parthanatos-related genes were gathered from previous publications, manual curation, and the GeneCards platform [1,2,15,16,17]. Ultimately, we retrieved 40 parthanatos-related genes for subsequent in-depth analysis.

### 2.2. Gene Expression Analysis

The mRNA expression profiles of parthanatos-related genes of breast cancers and corresponding normal breast tissues were downloaded from TCGA and the Genotype-Tissue Expression (GTEx) databases. R software (v 4.0.2) was utilized for data analysis, and the R package “ggplot2” (v 3.3.3) was employed for visualization. Wilcoxon tests were conducted to analyze significant differences. The Human Protein Atlas (HPA) database (https://www.proteinatlas.org/, accessed on 1 February 2025) was used to assess the protein expression differences in parthanatos-related genes between breast cancer tissues and normal mammary tissues.

### 2.3. Analysis of the Correlation and Protein–Protein Interactions (PPI) Between Parthanatos and Other Forms of Cell Death

Other forms of cell death encompassed 12 other PCD modalities, including apoptosis, pyroptosis, ferroptosis, autophagy, entotic cell death, cuproptosis, necroptosis, lysosome-dependent cell death, PANoptosis, and oxeiptosis. A total of 815 other PCD-related genes were curated from various sources, including GSEA gene sets, GeneCards, relevant review articles, and manual compilation [1,18].

To assess the correlation between parthanatos and other PCD patterns, we used the single-sample gene set enrichment analysis (ssGSEA) algorithm to score parthanatos-related genes alongside other PCD gene sets within the mRNA expression profiles from TCGA [19]. Subsequently, the Spearman correlation method was employed to examine the correlation between the ssGSEA scores of parthanatos and those of other PCD pathways. The results were visualized using the ggplot2 package in R.

Additionally, we explored the PPIs of parthanatos and other PCD patterns with a Spearman R-value greater than 0.3, utilizing the STRING database for PPI network analysis [20]. From these interactions, we selected high-confidence pairs for further analysis. The PPI network was visualized using the igraph package in R, producing a network map depicting the PPIs between parthanatos and other PCD pathways.

### 2.4. Single-Cell Sequencing Data Analysis

The single-cell breast cancer dataset is derived from GSE163558, which includes 8 TNBC, 13 ER/PR+HER2-, and 6 HER2+ cases [13]. To ensure quality control of the raw count data, we established the following criteria: (A) min.cells = 3, min.features = 200; (B) nCount_RNA ≥ 1000; (C) 200 ≤ nFeature RNA ≤ 8000; (D) percent.mt ≤ 20. To address the issue of varying total counts among cells, we applied a global scaling normalization method (LogNormalize) to normalize the data. This involved normalizing the feature expression measurements of each cell by their total expression, followed by multiplication by a scaling factor (10,000), and finally, transforming the values using the natural logarithm. Based on the “vst” algorithm, we identified feature genes that exhibited high intercellular differences in the dataset and normalized all genes for further analysis.

To simplify computations and remove data noise, we performed a joint analysis using PCA and Harmony for batch correction and dimensionality reduction. Harmony applies principal component analysis to embed the transcriptomic expression profile into a low-dimensional space and then iteratively removes dataset-specific effects. Based on the work of Pal et al., we manually annotated the single-cell data after quality control, identifying a total of 8 cell types, including normal epithelial cells, tumor epithelial cells, proliferation tumor epithelial cells, fibroblasts, endothelial cells, T cells, B cells, and macrophage cells [13]. We visualized these cell subgroups using t-distributed stochastic neighbor embedding (tSNE) [21]. Additionally, we employed the ssGSEA algorithm to score the parthanatos-related gene set within the single-cell dataset [19].

### 2.5. Unsupervised Clustering Based on Parthanatos-Related Genes

We randomly divided the bulk RNA-seq data of 1082 samples from the TCGA-BRCA cohort into training (60%) and validation (40%) sets. Unsupervised clustering analysis was performed using the “ConsensusClusterPlus” R package developed by Wilkerson et al. [22], with the following specific parameters: reps = 1000, pItem = 0.8, clusterAlg = “km”, distance = “Euclidean”, maxK = 6. The optimal number of clusters was determined based on the consensus cumulative distribution function and delta area plot. A heatmap was generated to depict the expression characteristics of each parthanatos-related gene in this molecular classifier.

### 2.6. Comparison of Clinical, Genomic, and Transcriptomic Features of Parthanatos-Related Breast Cancer Subtypes

To explore the significance of this classifier, we assessed clinicopathological features, prognosis, genomic landscape, biological functions, and immune microenvironment characteristics between different BC subtypes. We calculated the *p*-values for differences in clinicopathological features between subtypes using SPSS 20.0 software. The survival curves for prognosis comparison were plotted using the R package “survival” (v 3.3.1). Mutation waterfall plots were generated with the “maftools” package in R [23], while gene set enrichment analysis (GSEA) was employed to compare biological functions across subtypes [24]. Finally, we utilized tools such as “ESTIMATE”, “ssGSEA”, and “xCELL” to analyze the immune microenvironment characteristics [25,26].

### 2.7. Identification of Potentially Sensitive Drugs for BC Patients Based on Parthanatos Molecular Classifier

The “oncoPredict” R package was used to predict the response of 1082 BC patients to chemotherapeutic agents [27]. This R package integrates in vitro and in vivo drug screening, enabling the prediction of tumor response to a large number of drugs screened in cancer cell lines. Ultimately, we analyzed the differences in drug response among parthanatos subtypes of BC patients.

### 2.8. Parthanatos-Related Breast Cancer Subtypes and Response to PARP Inhibitors

We utilized the preprocessed expression data and corresponding clinical data of the PARP inhibitor arm (*N* = 71) from the I-SPY2 study [14]. To develop the classification model, logistic regression analysis was performed using the “caret” R package, with training control established by 10-fold cross-validation for robust model evaluation. The model was trained using a generalized linear model (GLM) with a binomial family to classify breast cancer subtypes based on parthanatos-related gene expression. This classifier was applied to the I-SPY2 PARP arm expression data to classify BC based on parthanatos gene expression.

### 2.9. Statistical Analysis

Categorical variables were compared using the Chi-Square test or the Fisher exact test where appropriate. Continuous variables were tested with a *t*-test or Wilcoxon rank-sum test, where appropriate. Survival was estimated using the Kaplan–Meier method, which is commonly used to assess the time to event data, such as survival or disease progression. Overall survival (OS) was defined as the time from diagnosis or treatment initiation to death from any cause, while progression-free survival (PFS) was defined as the time from diagnosis or treatment initiation to disease progression or death, whichever occurred first. A log rank test was used to determine whether a factor was associated with survival. Two-sided *p*-values less than 0.05 were considered to be statistically significant. All analyses were performed using SPSS 20.0 software.

## 3. Results

### 3.1. Differential Expression of Parthanatos-Related Genes in Breast Cancer and Normal Breast Tissue

We initiated our analysis by examining mRNA expression of parthanatos-related genes in breast cancers and normal breast tissue. Paired and unpaired analysis was conducted by combining TCGA and GTEx transcriptome datasets. The results revealed that the majority of parthanatos-related genes showed significant differences between tumor and normal tissue (Figure 1A,B). Among these, 20 genes, such as AIFM1, MIF, and PARP1, exhibited a significant increase in tumors compared with normal tissues, while 15 genes, including PTEN, NAMPT, and GAS5, demonstrated a marked reduction in expression (Figure 1A,B). Immunohistochemical results obtained from the HPA database further corroborated these findings at the protein level, aligning with the observed transcriptional differences (Figure 1C). These results suggest that parthanatos might be associated with the development of breast cancer.

Next, we utilized genomic data from TCGA, including 1082 breast cancer patients, to depict the mutation frequencies and types of parthanatos-related genes in breast cancer. Our analysis revealed that parthanatos-related genes exhibited a low mutation frequency in breast cancer, whereas CNVs were relatively high (Figure 2A,B). The gene with the highest frequency of CNVs was MCL1 (amplification, 11%), followed by PARP1 (amplification, 8%), TOMM20 (amplification, 7%), GAS5 (amplification, 5%), and PTEN (deletion, 5%) (Figure 2B). Additionally, genes with CNV amplification, such as PARP1 and TOMM20, were also found to be upregulated in breast cancer, while genes with CNV deletion, such as PTEN, were downregulated. Further analysis confirmed that these CNVs significantly impacted the mRNA expression levels of corresponding genes, suggesting that CNV is one of the mechanisms driving the differential expression of parthanatos-related genes in breast cancer (Figure 2C).

### 3.2. Correlation Between Parthanatos and Other Forms of Cell Death in Breast Cancer

We analyzed the correlation between parthanatos and 12 other well-characterized forms of cell death. Our results showed that the parthanatos pathway score was positively correlated with most forms of cell death, including apoptosis (R = 0.398, *p* < 0.001), ferroptosis (R = 0.322, *p* < 0.001), entosis (R = 0.329, *p* < 0.001), and lysosome-dependent cell death (LCD) (R = 0.320, *p* < 0.001), while it was negatively correlated with NETosis (R = −0.196, *p* < 0.001) (Figure 3A). Additionally, we analyzed the PPI networks between the parthanatos pathway and other programmed cell death pathways. Our findings revealed that parthanatos-related proteins exhibit extensive interactions with proteins associated with other forms of cell death (Figure 3B).

### 3.3. Single-Cell Expression Pattern of Parthanatos-Related Genes

To investigate the expression levels of parthanatos-related genes among various cell types in breast cancer at the single-cell level, we analyzed the single-cell dataset GSE163558. We identified and annotated eight distinct cell subgroups via tSNE analysis and applied the ssGSEA algorithm to score the parthanatos-related gene set (Figure 4A–C). We found that cancer cells were the primary cell type exhibiting a high parthanatos score across different molecular subtypes (Figure 4D–F). Notably, tumor epithelial cells exhibited a significantly higher parthanatos score compared to normal epithelial cells, indicating that cancer cells were the main contributors to parthanatos in tumor tissues (Figure 4D–F). Furthermore, in the ER/PR+ and HER2+ subtypes, macrophages also demonstrated a relatively high parthanatos score, suggesting a potential relationship between parthanatos and the immune microenvironment (Figure 4D–F).

Given that DNA damage is a known trigger of parthanatos [28], we investigated whether the genomic instability level in breast cancer cells correlates with parthanatos activation at the single-cell level. Thus, we employed the inferCNV R package to quantify genomic instability scores in cancer cells, using normal cells as a reference. Our analysis revealed a positive correlation between the genomic instability score of cancer cells and the parthanatos score (Figure 4G–I), indicating a potential association between the parthanatos score and genomic instability [28].

### 3.4. Defining the Molecular Subtypes of Breast Cancers Based on Parthanatos Activity

To determine whether parthanatos is an intrinsic biological characteristic that can distinguish different subtypes of breast cancer, we performed consensus clustering on 1082 samples with transcriptomic data through random resampling (1000 iterations). Based on the “elbow” point in the relative change in the area under the consensus distribution function curve, we selected two as the final number of clusters (Appendix A). Using unsupervised clustering with parthanatos-related genes, we divided the 1082 samples into two distinct subtypes: C1 and C2. The distribution of samples across the subtypes was as follows: C1, 46.5% (*N* = 503) and C2, 53.5% (*N* = 579). Both the training and validation sets demonstrated that this classification was stable and reproducible. The fundamental difference between the two subtypes lay in the higher parthanatos activity observed in C1, where the majority of genes, such as MCL1, TOMM20, SQSTM1, DDB1, and PARP1, showed higher expression levels compared to C2 (Figure 5A–C). Due to NAD+ depletion being a key biomarker of cell death, we compared the expression of key genes involved in this process between the C1 and C2 subtypes. Our analysis revealed that C1 exhibited higher expression of CD38, PARP1, and SIRT1, while NAMPT was downregulated compared with C2 (Figure 5D), supporting a greater extent of NAD+ depletion and cell death in C1. We then performed pathway enrichment analysis on the differentially expressed genes within the parthanatos-related gene set between C1 and C2. Our analysis revealed that parthanatos-related genes upregulated in C1 were primarily enriched in DNA damage repair pathways (Figure 5E), whereas those upregulated in C2 were predominantly associated with immune-related pathways (Figure 5F).

We further analyzed the correlation between these subtypes and their clinicopathological and prognostic features. No significant differences were shown in clinicopathological features between the two subtypes, such as ethnicity, pathological type, ER/PR/HER2 status, tumor size, lymph node metastases, distant metastases, and stage (Appendix A). However, we found that the C1 subtype exhibited significantly higher mRNA levels of MKI67 compared to the C2 subtype (median: 1915 vs. 1546, *p* < 0.001) (Appendix A), indicating a potential association between parthanatos activation and higher proliferative activity. In addition, Kaplan–Meier survival analysis revealed that the C1 subtype was associated with worse overall survival (unadjusted HR = 0.69, 95% CI: 0.48–1.00; log-rank *p* = 0.05), but no significant difference in progression-free survival was observed between the two groups (unadjusted HR = 0.82, 95% CI: 0.58–1.16; log-rank *p* = 0.27). These findings suggest that high parthanatos pathway activity may be associated with poorer long-term but not short-term outcomes in breast cancer patients (Figure 5G,H).

Then, we compared parthanatos pathway activation levels across different breast cancer subtypes and found that HER2-positive breast cancer exhibited significantly higher parthanatos pathway activation compared to TNBC and ER/PR + HER2- breast cancer (Appendix A). We conducted a subgroup analysis stratified by molecular subtype. C1 exhibited significantly higher parthanatos pathway activation than C2, regardless of the subtype (Appendix A). Moreover, across all subtypes, C1 showed upregulation of various DNA damage repair pathways compared to C2, suggesting that the parthanatos-based classification (C1 vs. C2) is stable across different breast cancer subtypes (Appendix A). Finally, we explored prognostic differences and immune activity levels between C1 and C2 across different subtypes. In ER/PR + HER2- and HER2+ breast cancers, C1 tended to be associated with a worse prognosis compared to C2, and C1 exhibited lower immune scores than C2 (Appendix A). However, in TNBC, no significant differences were observed between C1 and C2 in terms of prognosis or immune scores (Appendix A).

We then characterized the genomic features of each subtype. Despite little difference in the somatic mutation profiles (Appendix A), the CNV patterns of the two subtypes were notably distinct. C1 exhibited significantly higher copy number amplifications in several genes compared to C2, including MCL1, SETDB1, TOMM20, SMYD3, AKT3, NOTCH2, SPRTN, and PARP1 (Appendix A). Pathway enrichment analysis indicated that these genes play critical roles in pathways such as apoptosis and protein autoprocessing. Given the synthetic lethality effect between *BRCA* mutations and PARP1, a key gene in the parthanatos pathway, we further investigated the interaction between *BRCA* mutations and parthanatos-based subtypes in the TCGA dataset. As expected, PARP1 expression, DNA damage signaling, and the base excision repair pathway (a DNA repair pathway closely associated with PARP1 function) were significantly upregulated in *BRCA*-mutant breast cancers compared to *BRCA*-wildtype cases (Appendix A), suggesting that genomic instability associated with *BRCA* mutations may contribute to PARP1 activation. However, we observed no significant difference in parthanatos pathway scores between *BRCA*-mutant and *BRCA*-wildtype breast cancers (Appendix A). Similarly, the proportion of *BRCA*-mutant cases between the two parthanatos-based subtypes showed no difference (C1: 4.77% vs. C2: 3.80%, *p* = 0.429) (Appendix A). These findings suggest that the extent of DNA damage in *BRCA*-mutant breast cancers may not be sufficient to induce overactivation of PARP1, thereby failing to trigger parthanatos activation.

### 3.5. Comparison of Biological Characteristics and Immune Microenvironment Between Parthanatos-Related Breast Cancer Subtypes

We identified 438 differentially expressed genes (DEGs) between C1 and C2 (log2FC > 0.5, padj < 0.05) (Figure 6A). Pathway enrichment analysis of these DEGs revealed upregulation of various DNA damage repair pathways, which aligns with the high expression of parthanatos-related genes in the C1 subtype. Additionally, pathways related to extracellular matrix degradation and gliogenesis were significantly downregulated, suggesting that C2 is more closely associated with the stromal microenvironment (Figure 6B).

We also compared the immune microenvironment between the two subtypes. The ImmuneScore calculated by ESTIMATE in C2 were higher than those in C1 (Figure 6C). Specifically, analyses using ssGSEA and xCell tools consistently showed that the abundances of M1 macrophages and B cells were significantly higher in C2 compared to C1 (Figure 6D,E). This immune activation may partially explain the favorable prognosis associated with the C2 subtype.

### 3.6. Relationship Between Parthanatos BC Subtypes and Response to PARP Inhibitors

To explore differences in drug sensitivity between the two parthanatos BC subtypes, we used cell line drug screening datasets and the “oncoPredict” package [27]. Drug sensitivity analysis was performed using TCGA transcriptomic data. The analysis utilized “oncoPredict” package to predict the sensitivity of TCGA samples to specific drugs, leveraging drug sensitivity data from over 300 drugs and expression profiles from more than 1000 human cancer cell lines provided by GDSC2 [29]. A drug sensitivity score was calculated for each sample, where a lower score indicates higher sensitivity to the drug. We found that C2 was more sensitive to Cisplatin, Oxaliplatin, Epirubicin, and Paclitaxel compared to C1 (Figure 7A).

The I-SPY2 PARP inhibitor treatment cohort included 71 primary breast cancer patients who received the treatment of neoadjuvant chemotherapy plus a PARP inhibitor. Based on the expression data of core-needle biopsy tumor samples before neoadjuvant therapy, these samples were classified into 12 C1 subtypes and 59 C2 subtypes. We found that the pCR (pathologic complete response) rate (2/12 vs. 25/59, *p* = 0.086) in the C1 subtype was lower than that in C2, indicating that C1 may predict treatment resistance to PARP inhibitors (Figure 7B,C). The immune microenvironment plays a critical role in treatment response. We analyzed immune scores in the I-SPY dataset and found that C1 had significantly lower immune scores than C2, which is consistent with our findings in the TCGA dataset. To explore the potential influence of the immune score in PARP inhibitor response, we performed univariate logistic analysis to assess the effect of the immune score and parthanatos-based classification on pCR. Our results showed that parthanatos-based subtypes (unadjusted OR = 3.68, 95% CI 0.87–25.31, *p* = 0.11) and immune score (unadjusted OR = 2.24, 95% CI 0.85–6.14, *p* = 0.27) were both associated with pCR receiving PARP inhibitors. After adjusting for the immune score, the parthanatos-based subtypes still had a trend associated with pCR (adjusted OR = 2.67, 95% CI 0.53–20.18, *p* = 0.11).

## 4. Discussion

This study represents the first to investigate the role of parthanatos in breast cancer. We observed significant differential expression of parthanatos-related genes between breast cancer and normal breast tissues. CNVs of parthanatos-related genes were highly prevalent in breast cancer and significantly impacted gene expression levels. At the single-cell level, parthanatos-related genes were primarily expressed in breast cancer cells and macrophages, with genomic instability scores in cancer cells showing a positive correlation with parthanatos pathway scores. Unsupervised clustering analysis revealed that breast cancer can be stably classified into two subtypes based on the expression profiles of parthanatos-related genes: C1 (parthanatos-high) and C2 (parthanatos-low). Compared to C2, C1 exhibited worse prognosis, reduced immune infiltration, and potential resistance to PARP inhibitors in real-world clinical cohorts.

Previous studies reported that parthanatos is associated with the development and progression of ovarian, lung, and gastric cancers [3,30,31]. In breast cancer, research has been largely limited to analyzing the expression status and clinical significance of the PARP1 gene [32]. This may be attributed to the fact that PARP1 has predominantly been reported to play a key role in the DNA damage response and the repair of single-stranded breaks (SSBs) through the base excision repair (BER) in breast cancer [33] rather than as a critical gene in the parthanatos. In this study, through the integration of genomic and transcriptomic data, we found that the majority of parthanatos-related genes showed significant differences between breast tumor and normal mammary tissues. Additionally, parthanatos-related genes displayed widespread CNVs in breast cancer, which likely contributed to changes in mRNA expression levels. At the single-cell level, parthanatos-related genes were mainly expressed in tumor cells, with genomic instability scores in tumor cells showing a positive correlation with parthanatos pathway scores. This is consistent with previous mechanistic studies showing that parthanatos are primarily induced by excessive DNA damage [34].

We further investigated the clinical significance of parthanatos in breast cancer. We found that higher parthanatos activity was associated with poor prognosis in breast cancer. Consistent with this finding, studies have indicated that higher PARP1 expression predicts worse clinical outcomes in breast cancer [35,36]. However, high parthanatos activity has been associated with better prognosis in cancers such as gastric cancer and glioma [15,37]. This discrepancy in survival may be explained by differences in cancer types and the dual-edged nature of parthanatos in tumor biology [4]. Additionally, we found that lower parthanatos activity in breast cancer was associated with immune microenvironment activation. Similar findings have been reported in gastric cancer, where subtypes with lower parthanatos expression exhibit a more active immune microenvironment [15]. Moreover, former studies suggest that PARP inhibitors can enhance the anti-tumor immune response of TNBCs, which has been associated with improved prognosis [38,39,40,41]. This finding aligns with our study, where the BC subtype with low parthanatos expression exhibited immune microenvironment activation and was associated with a better prognosis.

In our study, the C1 subtype, characterized by high expression of parthanatos-related genes, was associated with poor response to PARP inhibitors in breast cancer. Extensive DNA damage leads to the overactivation of PARP1, which serves as the initiating step of parthanatos, followed by the translocation of apoptosis-inducing factor and DNA fragmentation, ultimately resulting in cell death [1,5]. Interestingly, PARP inhibitors exert their anticancer effects by inhibiting PARP1 enzymatic activity, thereby impairing the base excision repair pathway. This mechanism induces synthetic lethality in *BRCA*-mutant tumors, which are already deficient in homologous recombination repair (PARP inhibitors: synthetic lethality in the clinic). We hypothesize that parthanatos activation may render cancer cells more susceptible to death. However, upon PARP inhibitor treatment, the overactivated PARP1 is inhibited, preventing the sustained induction of parthanatos-mediated cell death. As a result, cancer cells may continue to survive, leading to resistance to PARP inhibitors. Notably, a recent study reported that in *BRCA*-mutant breast cancers, PARP1 amplifications occur significantly more frequently in post-treatment recurrent lesions than in primary tumors [8], suggesting that PARP1 amplifications may contribute to post-treatment recurrence in *BRCA*-mutant breast cancer. Together with these findings, our results suggest that overexpression of parthanatos-related genes may mediate resistance to PARP inhibitors and contribute to poor prognosis in breast cancer. Further investigations are warranted to elucidate the underlying molecular mechanisms.

PARP inhibitors (PARPi) are the first clinically approved drugs designed to exploit synthetic lethality, which have been approved for adjuvant therapy in early stage *BRCA*-mutated breast cancer and salvage therapy in advanced *BRCA*-mutated breast cancer [42,43,44]. Additionally, multiple clinical trials, such as PARTNER [45], are now exploring their potential application in breast cancers without *BRCA* mutations. However, it is important to recognize that a considerable proportion of patients do not benefit from PARP inhibitors [46,47,48]. Our findings suggest that parthanatos activation may serve as a potential biomarker of resistance to PARP inhibitors in breast cancer, which could help refine patient selection and enable more precise identification of individuals likely to benefit from PARP inhibitor therapy.

Our study has certain limitations, primarily due to its reliance on bioinformatics analysis alone. Further in vivo and in vitro experiments are required to elucidate the mechanisms underlying the role of parthanatos in BC and to validate our conclusions. Additionally, identifying effective predictive biomarkers for prognosis and treatment response remains a significant challenge. Larger cohorts of patients receiving PARP inhibitor therapy are needed to further validate the association between parthanatos and the therapeutic efficacy of PARP inhibitors in breast cancer.

## 5. Conclusions

The study systematically analyzed the role and clinical significance of parthanatos in breast cancer. We observed significant differential expression of parthanatos-related genes between breast cancer and normal breast tissues. CNVs of parthanatos-related genes were highly prevalent in breast cancer and significantly impacted gene expression levels. At the single-cell level, parthanatos-related genes were primarily expressed in breast cancer cells and macrophages, with genomic instability scores in cancer cells showing a strong positive correlation with parthanatos pathway scores. Using unsupervised clustering, we classified samples into two subtypes: C1 (parthanatos-high) and C2 (parthanatos-low). Compared to C2, C1 exhibited a worse prognosis, reduced immune infiltration, and potential resistance to PARP inhibitors in real-world clinical cohorts. Our study enhances the understanding of the role of parthanatos in the development and progression of breast cancer and highlights its potential association with PARP inhibitor resistance and poor prognosis.

## Figures and Tables

**Figure 1 bioengineering-12-00586-f001:**
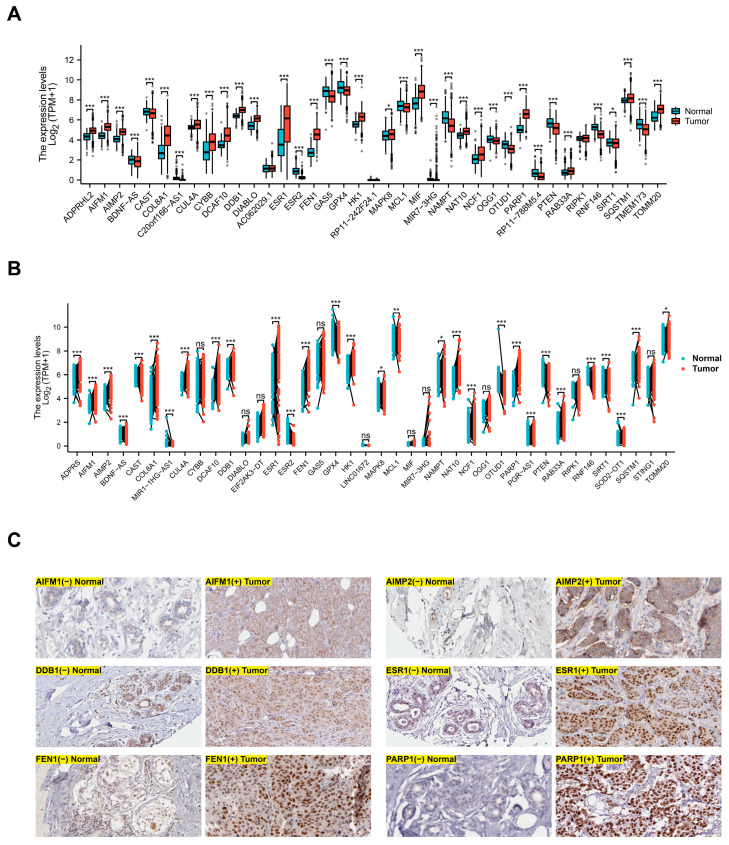
Expression analysis of parthanatos-related genes in breast cancer. (**A**) mRNA expression levels of parthanatos-related genes in normal breast tissues and breast cancer from TCGA and GTEx databases. (**B**) Paired analysis of parthanatos-related gene mRNA expression in paracancerous tissues and breast cancer from the TCGA database. (**C**) Protein expression of the parthanatos-related gene in breast cancer and normal breast tissues from the Human Protein Atlas database. * *p* < 0.05, ** *p* < 0.01, and *** *p* < 0.001. ns: not significant.

**Figure 2 bioengineering-12-00586-f002:**
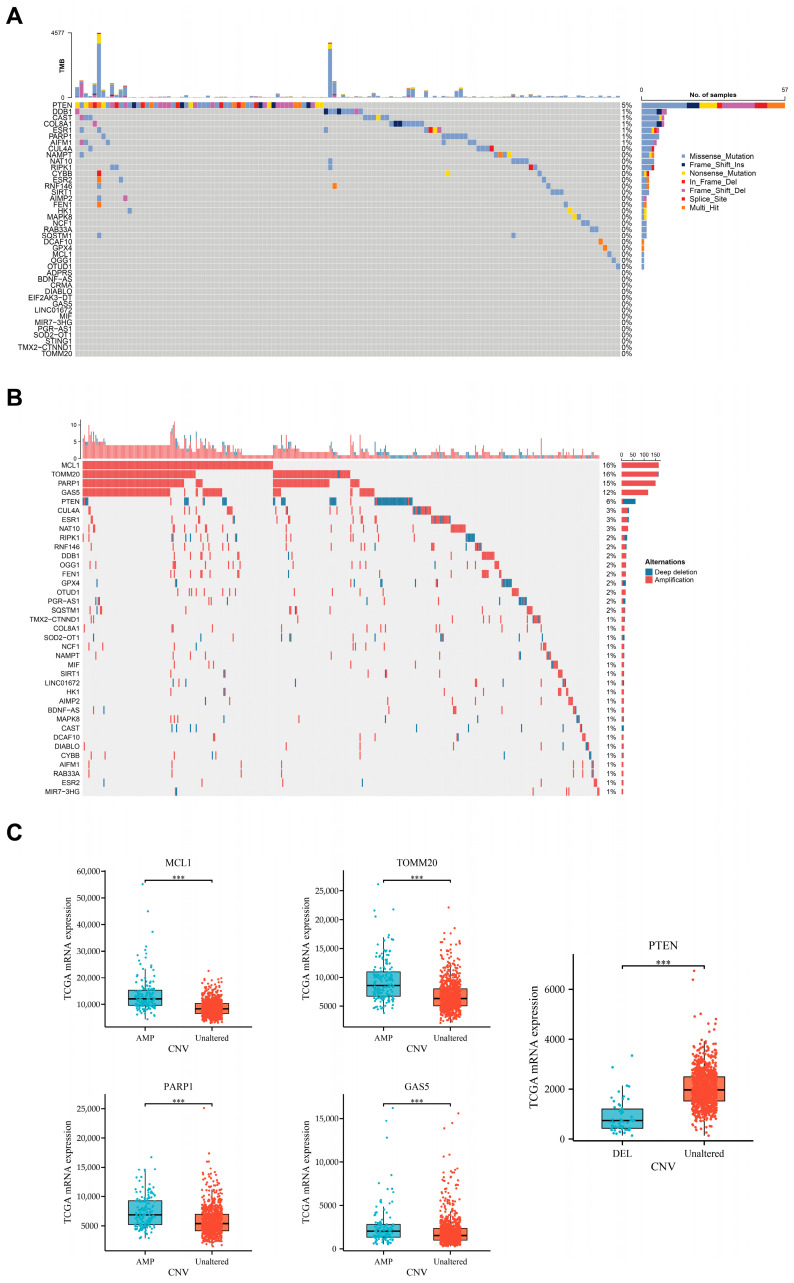
Genomic alterations of parthanatos-related genes in breast cancer. (**A**) Mutation profiles of parthanatos-related genes. Genes are ordered by mutation rate; TMB (mut/Mb) is recorded on the top of the plot. (**B**) Copy number variation (CNV) profiles of parthanatos-related genes. (**C**) mRNA expression levels of parthanatos-related genes in groups with or without CNVs. Abbreviations: AMP, amplication; DEL, deletion. *** *p* < 0.001.

**Figure 3 bioengineering-12-00586-f003:**
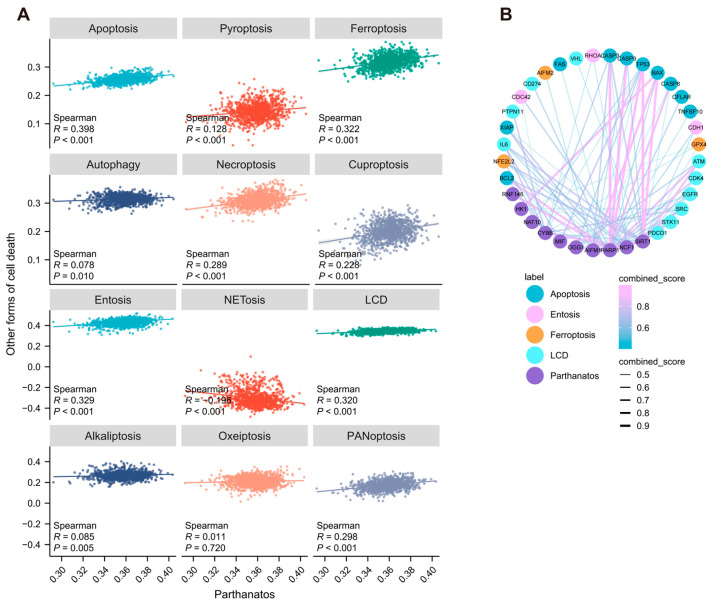
Correlation between parthanatos and other forms of cell death in breast cancer. (**A**) Correlation analysis between parthanatos pathway scores and other forms of cell death pathway scores using data from TCGA. The Pearson correlation coefficient R between 0–0.3 indicates no correlation or weak correlation, 0.3–0.5 indicates a weak correlation, 0.5–0.8 indicates a moderate correlation, and 0.8–1.0 indicates a strong correlation. (**B**) PPI network between the parthanatos pathway and other forms of cell death pathway. The combined score represents the overall strength of protein–protein interactions. Abbreviations: LCD, lysosome-dependent cell death.

**Figure 4 bioengineering-12-00586-f004:**
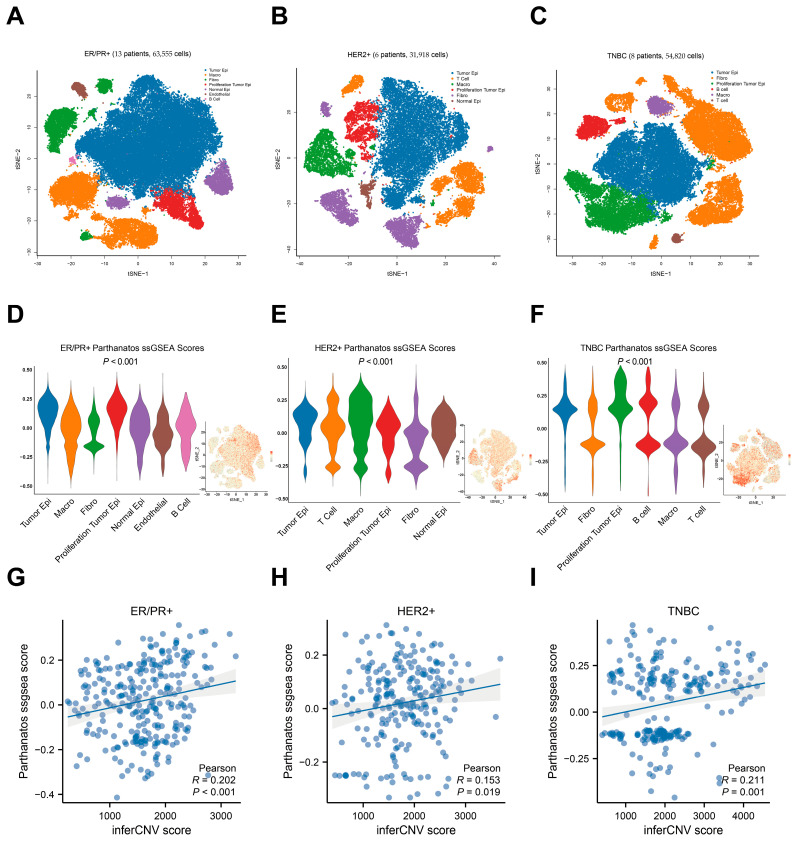
Single-cell expression of parthanatos-related genes. (**A**–**C**) t-SNE plots of combined scRNA-seq profiles of total cells from 13 ER+ tumors, 6 HER2+ tumors, and 8 TNBC tumors, respectively. (**D**–**F**) Violin plot displaying parthanatos scores across breast cancer subtypes, with a miniature t-SNE plot positioned in the bottom right corner of the violin plot. The *p*-values were calculated by the Kruskal–Wallis test. (**G**–**I**) Correlation analysis between inferCNV and parthanatos scores in various breast cancer subtypes. Abbreviations: Epi, epithelial; Fibro, fibroblast; Macro, macrophage.

**Figure 5 bioengineering-12-00586-f005:**
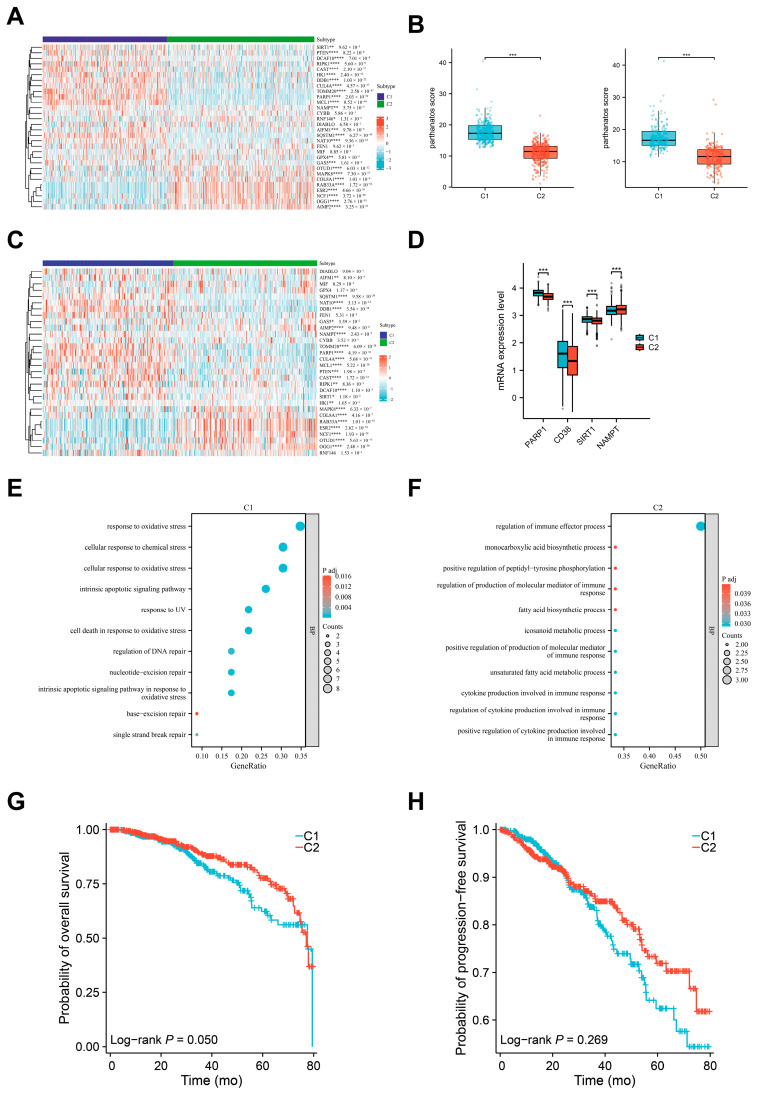
Parthanatos-based clustering analysis. (**A**) Expression difference in parthanatos-related genes between C1 and C2 subtypes in training sets. (**B**) Box plots showing parthanatos pathway scores for two clusters (C1 and C2) in training (left) and validation (right) sets. (**C**) Expression difference in parthanatos-related genes between C1 and C2 subtypes in validation sets. (**D**) mRNA expression levels of key genes involved in NAD+ depletion process between C1 and C2 subtypes. (**E**) Pathway enrichment analysis on parthanatos-related genes upregulated in the C1 subtype. (**F**) Pathway enrichment analysis on parthanatos-related genes upregulated in the C2 subtype. (**G**,**H**) Kaplan–Meier survival analyses of C1 and C2 subtypes. Abbreviations: Mo, months. * *p* < 0.05, ** *p* < 0.01, *** *p* < 0.001 and **** *p* < 0.0001.

**Figure 6 bioengineering-12-00586-f006:**
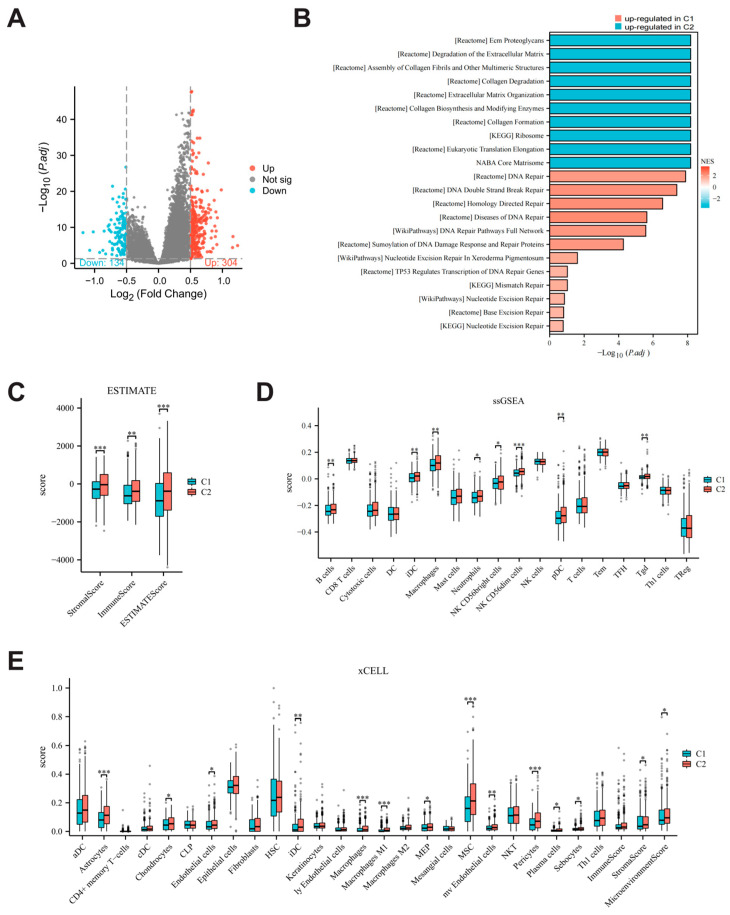
Biological characteristics and immune microenvironment in parthanatos-based BC subtypes. (**A**) Volcano plot showing the distribution of differentially expressed genes (DEGs) between C1 and C2. Genes with significant differential expression (padj < 0.05, Log2FC > 0.5) are highlighted, with upregulated genes in red and downregulated genes in blue. Non-significant genes are shown in gray. (**B**) Gene set enrichment analysis (GSEA) between C1 and C2 subtypes. (**C**–**E**) Differences between the two clusters in ESTIMATE, ssGSEA, and xCELL algorithms. Abbreviations: NES, normalized enrichment score; Up, upregulated; Down, downregulated; Not sig, not significant. * *p* < 0.05, ** *p* < 0.01 and *** *p* < 0.001.

**Figure 7 bioengineering-12-00586-f007:**
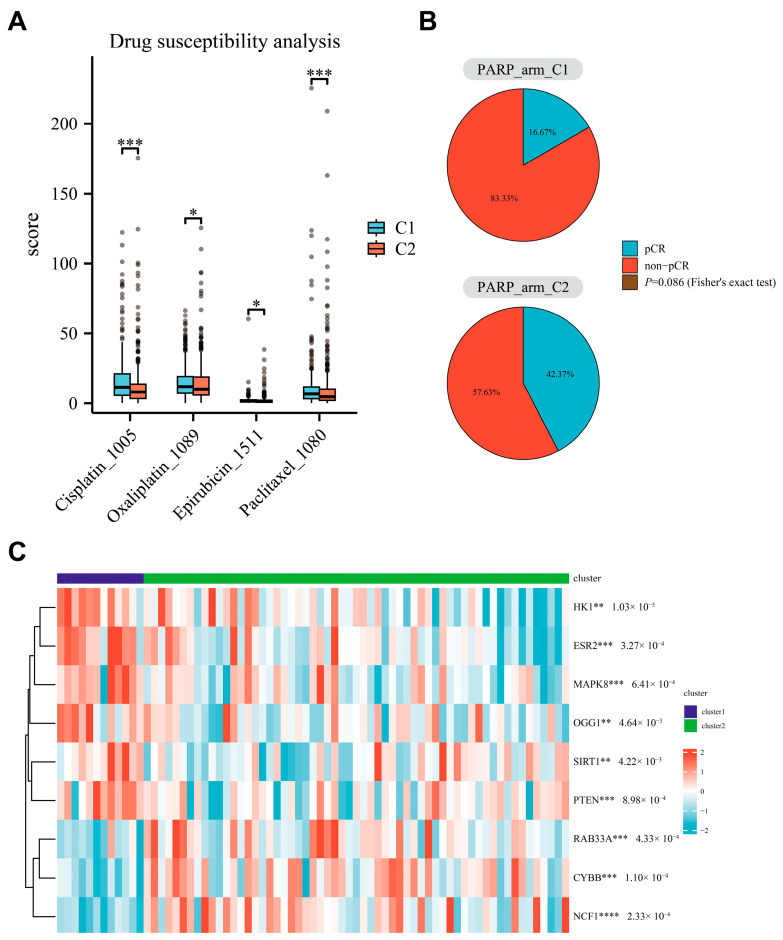
Response to PARP inhibitors in parthanatos-based BC subtypes. (**A**) Drug susceptibility analysis of common chemotherapeutic agents in breast cancer. (**B**) Efficacy of PARP inhibitors in C1 and C2 subtypes, with *p*-values calculated using Fisher’s exact test. (**C**) Expression difference in parthanatos-related genes between C1 and C2 subtypes in the I-SPY2 PARP inhibitor treatment cohort, with *p*-values calculated using Wilcoxon rank-sum test. Abbreviations: pCR, pathologic complete response. * *p* < 0.05, ** *p* < 0.01, *** *p* < 0.001 and **** *p* < 0.0001.

## Data Availability

All the genomic, transcriptome and clinic data used in this study were obtained from public datasets described in the Section 2. Any additional information required is available from the corresponding author upon reasonable request.

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
