# Peer review of "Characterization of Parthanatos in Breast Cancer: Implications for Prognosis and PARP Inhibitor Resistance"

_bioengineering, 2025, doi:10.3390/bioengineering12060586_

Round 1
Reviewer 1 Report
Comments and Suggestions for Authors
This study is a novel to demonstrate the role of parthanatos in breast cancer whereas there have been more studies in other cancer types. Authors analyzed multiomics data from large cohorts in TCGA and well described differential status in subtypes of parthanatos. The study defined two distinct groups in breast cancer related with parthanatos using mRNA, CNV mutation and single cell RNAseq. Subtypes were well defined in the study with C1: high DNA damage repair pathway related genes and C2: stromal microenvironment related genes. Further, the study clinically claimed C1 subtype exhibited poor prognosis and potential resistance after PARPi inhibition due to high activity of parthanatos although the reasoning still needs to further investigate
Specific comments
- need to show accurate pathway score number comparison ssGSEA in tumor and macrophage subtypes (fig 4D-F) due to low resolution, and hard to read in the figure (not sure if figures were embedded temporally in this printed version)
- need possible description or to explain why C1 subtype was marginally linked to OS but not in progression-free progression (fig5G-H)
- It was not sufficiently explained why only HER2+ cohort showed meaningful OS (figS2D) any possible reasons can be shown? and it was also hard to tell clear correlation with immune score since it similarly showed higher immune scores in ER+ but no significant difference in OS (figS2E)
- Authors claimed BRCA mutation status may contribute to PARP1 activation but not Parthanatos pathway since the extent of DNA damage in BRCA-mutated BCC might not be sufficient to overactivate PARP1 failed to trigger Parthanatos activation. To emphasize this, DNA damage signaling pathway should be compared between BRCA wild type and mutation groups to confirm if its activation was higly correlated
Reviewer 2 Report
Comments and Suggestions for Authors
It would be interesting to know the reasons and types of non-corrispondence between tumor cells and this new marker, such as lymph nodes metastases, widespread blood diffusion, poor differentiation, chemotherapy resistence, high replication index, etc. There is an impact of Parthanatos on chemotherapy choice? The references about the mechanisms of this cells marker have to be updated.
